# Molecular Dynamics Study of Cellulose Nanofiber Alignment under an Electric Field

**DOI:** 10.3390/polym14091925

**Published:** 2022-05-09

**Authors:** Ruth M. Muthoka, Pooja S. Panicker, Jaehwan Kim

**Affiliations:** Creative Research Center for Nanocellulose Future Composites, Department of Mechanical Engineering, Inha University, 100 Inha-ro, Michuhol-ku, Incheon 22212, Korea; mwongelinruth@gmail.com (R.M.M.); pooja.panicker7@gmail.com (P.S.P.)

**Keywords:** cellulose, molecular dynamics, alignment, electric field

## Abstract

The alignment of cellulose by an electric field is an interesting subject for cellulose material processing and its applications. This paper reports an atomistic molecular dynamics simulation of the crystalline cellulose nanofiber (CNF) model in varying electric field directions and strengths. GROMACS software was used to study crystalline cellulose 1β consisting of 18 chains in an aqueous environment at room temperature, and an electric field was applied along the cellulose chain direction and the perpendicular direction with varying field strength. The root-mean-square displacement, radius of gyration, end-to-end length, and hydrogen bond population of the crystalline CNF model were analyzed to determine the effects of the applied electric field on the structure of the CNF model. The results suggest that the nanosecond electric field can induce the orientation of the CNF along the applied electric field direction. The alignment rate and ability to maintain the alignment depend on the electric field strength. Analysis of the radius of gyration, end-to-end length, and bond lengths for intrachain and interchain hydrogen bonds revealed no significant effect on the cellulose structure. Cellulose alignment in an electric field has the potential to broaden the design of electric field-induced processing techniques for cellulose filaments, thin films, and electro-active cellulose composites.

## 1. Introduction

Cellulose, the most abundant biopolymer produced in nature, has tremendous potential in many applications, including pulp, paper, additives, coatings, films, laminates, pharmaceuticals, foods, textiles, and composites [1,2]. Nano-sized cellulose, so-called nanocellulose, can be divided into cellulose nanocrystal (CNC) and cellulose nanofiber (CNF). Nanocellulose has unique characteristics, such as high strength and stiffness, low density, low thermal expansion coefficient, optical transparency, high hydrogen-bonding capacity, abundance, and renewable behavior.

The potential of nanocellulose is being utilized for structural composites, multifunctional composites, soft actuators, disposable sensors, biomedical devices, antimicrobial films, food packaging, separation/barrier membranes, and many more emerging applications [3,4,5,6]. Commercial exploitation of nanocellulose-based products has been made possible due to the ease of production of nanocellulose by chemical and mechanical methods [7,8]. Researchers have improved the mechanical properties by aligning the individual CNFs in orientations that are more ordered than random [9].

Several experimental works have been reported that demonstrated the effects of electric fields in the alignment of nanocellulose [9,10,11]. The experiments showed that the electric field applied can align CNC and CNF along the electric field direction. The native cellulose was aligned using an electric field application in organic solvents, chloroform, and cyclohexane [12]. The researchers reported the rotation of cellulose particles under an electric field of 200 V/mm and 1 kHz frequency.

CNC alignment in water under an AC electric field was reported, and increasing the electric field voltages by more than 2 kV/cm in a frequency range of 1 kHz to 2 MHz enhanced the alignment [13]. The alignment of regenerated cellulose chains by corona poling was reported with subsequent improvement in its piezoelectricity [11]. Ultrathin CNC films were aligned in electric-field-assisted shear assembly, in which the CNC orientation depended on both the electric field and frequency [14].

Nanofibrillated cellulose was aligned in silicon oil [15]. A permanent electric dipole moment in CNC dispersed in toluene and subjected to an electric field was reported [16]. Strong and tough nanocellulose filaments were made by aligning CNF under magnetic and electric fields [10]. Field-assisted alignment of CNF in a continuous flow system was recently reported with significant improvement in the mechanical properties [17].

Recently, improved piezoelectric properties of ultrathin CNF films were made by corona poling [18]. However, molecular aspects of the behavior of cellulose under an electric field and the corresponding effects on its structure remain unknown. Molecular dynamics (MD) simulations have been widely used to study crystalline cellulose [19]. MD simulation is a powerful tool for monitoring the atomic-level changes and better understanding crystalline cellulose’s effects and behavioral responses under electric fields. Cellulose MD simulations reported that cellulose surface wettability could be tuned by an applied electric field [20]. To the authors’ knowledge, MD simulations have never been studied for the alignment of a cellulose structure under an electric field.

In this work, atomistic molecular dynamics simulations were performed to study the alignment effect of crystalline CNF under various electric field strengths and the corresponding effects on its structure. The CNF model used in these simulations is crystalline cellulose 1β consisting of 18 chains in an aqueous environment and room temperature, and the GROMACS 2019 software was used for the MD simulation. The objective of this study is primarily to investigate the induced alignment of CNF along the direction of the applied electric fields. The changes in the physical, structural, and dynamic properties of the crystalline CNF model are inspected under the applied electric fields.

## 2. Methods

The atomic-scale MD simulations were conducted using a GPU-accelerated open-source Groningen Machine for Chemical Simulations (GROMACS) software (version 2019, developed at Stockholm Center for Biomembrane Research, Sweden) with All-Atom Optimized Potentials for Liquid Simulations (OPLS-AA) force field extended to parametrize cellulose molecules [21,22,23]. The TIP3P (Transferable Intermolecular Potential with three points) water model compatible with OPLS-AA was chosen to parametrize water molecules.

X-ray diffraction measurements and unit cell specification of crystalline cellulose 1*β* was used to build the simulation model [24]. Cellulose builder was used to making a 36-chain hexagonal cellulose model [25], which was then used to make an 18-chain model consisting of six sheets with overall zero charge. Each chain consisted of 20 glucose residues [19]. Periodic boundary conditions were applied in all there Cartesian directions of the simulation box. The cellulose structure consisting of 7614 atoms was placed at the center of a triclinic periodic box with *x*, *y*, and *z* box vectors of 19.842, 19.842, and 26.832 nm respectively, and an image distance, *d* of 8.0 large enough to allow the anticipated alignment, and 1,032,660 TIP3P water molecules were added for solvation.

The initial structures were first energy minimized using the steepest descent protocol for 100,000 steps to eliminate unfavorable clashes between atoms followed by two 0.5 ns equilibrations. The first equilibration was conducted for 0.5 ns in the constant temperature and constant volume (NVT) ensemble, followed by the second equilibration step for 0.5 ns in the constant temperature and constant pressure (NPT) ensemble. The energy minimization employed a maximum force value of 1000 kJ/nm/mol converging criterion.

The production MD was performed on the equilibrated starting structure in the NPT ensemble under 300 K temperature using the energy minimized cellulose coordinates as input. The velocity-rescale thermostat (relaxation time constant 0.1 ps) and the Parrinello-Rahman (relaxation time constant 1 ps) [26] with a grid spacing of 0.12 nm were used to maintain the pressure at 1 bar.

Short-range electrostatic interactions were cut off at 1.2 nm, and the long-range electrostatic interactions were calculated using the Particle Mesh Ewald algorithm [27]. All bond lengths were subjected to constraints using the Linear Constraint Solver (LINCS) algorithm [28]. A leapfrog algorithm was applied to integrate equations of motions for all simulations with a time step of 2 fs, and atomic coordinates were saved every 500 steps resulting in 10,000 trajectory frames.

The simulation was conducted in external oscillating electric fields acting on every atom in the system with varying electric field intensities from 0.001 mV/nm up to 0.2 mV/nm. This electric field intensity range was found from the real field strength: 1 V/mm to 200 V/mm. The electric field frequency for the MD simulation of the CNF model cannot be the same as the experimental frequency values since the simulation model size is in the nm range, and the simulation time should not be hours.

Thus, the electric field frequency and simulation time should be determined for the molecular level CNF model. In other words, in experimental electric field alignment, 1 kHz was chosen, which corresponds to the speed of electric field propagation is 10 m/s (=frequency × length = 1000 Hz × 0.001 m). Thus, the speed of electric field propagation should be the same as in the CNF model, and the frequency is 10 GHz (=10 m/s/1 nm). The maximum simulation time was chosen for 20 ns to emulate a static electric field.

The two simulation cases based on electric field application are shown in Figure 1. At first, the CNF model’s chain axis was offset by 10° from the z-axis (CASE 1, see Figure 1a). An electric field was applied along the z-axis to investigate the CNF orientation in the length direction (the y–z plane). The long duration of the simulation ensures that the model captures significant data for statistical analysis.

Analysis of the trajectories was conducted in addition to vis-à-vis zero-electric field conditions. The root-mean-square displacement (RMSD) of the cellulose structures at the time *t_o_* and *t* along the trajectory was analyzed to predict the arithmetical values of the deviations in the structure under different external electric fields. The radius of gyration, end-to-end length, and hydrogen bond disparity were evaluated using GROMACS software [21].

Visual Molecular Dynamics (VMD) software (University of Illinois, Urbana-Champaign, USA) [29] and PyMOL molecular graphics system were used to obtain snapshots of molecular structures [30]. The structural effect of the applied electric fields was investigated. Orientation of the CNF to the direction of the applied electric field was investigated. The orientation velocity, time taken for complete orientation, and the duration in which alignment is maintained for the different electric fields were calculated.

In CASE 2, the CNF model cross-section was tilted at 45° from the x-axis, and an electric field was applied to the cross-sectional direction (x-axis). Figure 1b shows the model configuration. The behavior of the cellulose structure under the electric field conditions was investigated at 1, 2, and 4.5 mV/nm. As in the previous case, vis-à-vis zero-electric field conditions in x-direction were conducted. To examine the structural denaturing caused by the electric field, RMSD, the radius of gyration, end-to-end length, and overall hydrogen bonds were investigated. VMD and PyMOL18 software were used to obtain snapshots of structural changes.

## 3. Results and Discussion

### 3.1. CASE 1: Electric Field along the Z-Axis

#### 3.1.1. Electric Field-Induced Alignment of Cellulose Structure

The simulation results clearly show the cellulose chains’ visible alignment parallel to the applied electric field direction. The alignment is induced at several picoseconds into the simulation, depending on the strength of the applied electric field. The degree of alignment regarding how quickly the alignment begins and how long full alignment is maintained depends on the applied electric field strength. In addition, it is important to examine if the alignment is lost later in the simulation.

Table 1 represents snapshots of the CNF alignment along the applied electric field direction along the z-axis. Upon application of the electric field, the CNF model is first observed to resist the external force exerted due to the applied electric field. At the lowest electric field (0.001 mV/nm), the original orientation angle is maintained throughout the simulation, with a slight twisting observed at 3 ns with no alignment observed.

The ability at which the cellulose structure could resist the applied force before induced alignment along the axis of the electric field reduced as the electric field intensity increased. A low electric field intensity of 0.005 mV/nm was observed to be sufficient to induce alignment of the cellulose parallel to the axis of the applied electric field. At this electric field, an alignment was induced after 1.35 ns in the simulation, with complete alignment achieved after 0.2 ns and maintained for only 0.5 ns. This suggests that low electric field intensity is insufficient to maintain the achieved alignment; hence, the CNF model reverts to its original orientation.

Increasing the electric field shows early onset of induced alignment of the CNF model and a significant increase in the time maintained the achieved alignment. At 0.06 mV/nm, the cellulose structure is observed to resist the electric field applied and maintains its initial orientation for 0.6 ns before alignment parallel to the axis of the applied electric field begins. This electric field is sufficient to maintain the achieved alignment throughout the simulation time up to 8 ns. This alignment result is similar to experimental results [10]. Increasing the electric fields further to 0.09 and 0.2 mV/nm only show rotations of the CNF model throughout the entire simulation.

This indicates that the corresponding force subjected to the CNF model is strong where induced rotations are dominant, thus, eliminating any possibility of alignment. Both the onset of alignment and the relative time the cellulose stayed aligned as a function of time for all the applied electric field strengths were analyzed as shown in Figure 2a,b. The onset of alignment of the CNF model to orient along the applied electric field is approximately 655 ps at 0.06 mV/nm electric field strength and 1350 ps at 0.005 mV/nm. As seen in Figure 2a, the onset of alignment is earlier as the electric field increases. Increasing the electric field has a higher chance of maintaining the achieved alignment, as shown in Figure 2b, with alignment maintained for 6.84 ns at 0.06 mV/nm.

The simulations were conducted twice to verify the reproducibility of these results. Figure 2c depicts the time for complete alignment and maintaining the achieved alignment at 0.06 mV/nm for repeat MD simulations referred to as 0.06 mV/nm_1 and 0.06 mV/nm_2. Appendix A shows snapshots of the reproducible alignment throughout the simulation at 0.06 mV/nm. It is clear from the graph that similar results were achieved and are accurately reproducible.

The onset of alignment, time taken to complete alignment, and the alignment-maintained time are similar. The simulations were extended further up to 28 ns (~three-times longer) to determine if the CNF recovered its lost alignment after 8 ns. Although mild rotations of the CNF were observed, no alignment was regained after being lost. These findings suggest that CNF can be aligned in the low electric field as recently reported [31].

Many studies have used electric field simulations in GROMACS using field frequency in the GHz range [32,33]. The instantaneous orientation in the strength of the low-frequency electric field does not happen as rapidly as in the strength of the high-frequency field. As a result, compared to the high-frequency electric field, GHz, the application of the low-frequency field below kHz resembles a static electric field.

A static electric field for electric poling is normally applied for more than 1 h to align CNF [1,9]. Within the simulation time frame of 10 ns, the systems with the high-frequency electric field spend more time experiencing instantaneous changes in the strength of the electric field. Consequently, the CNF rotation is maintained under a high-frequency electric field, such as that resulting from the poling effect of the static field.

#### 3.1.2. Dipole Moment

The behavior of the CNF’s overall dipole moment is important in indicating the associated alignment of the CNF due to the applied electric field. Many molecules possess an electric dipole moment that orients itself in the applied electric field direction. This study applies the electric field along the z-axis for CASE 1. Figure 3 shows the time evolution of the three components of the total dipole moment z, y, and x computed from the gromacs’s gmx dipoles, which compute the simulation systems’ total dipole and fluctuations [21]. A significant increase in the z-component of the dipole moment was observed with an increase in the electric field strength as shown in Figure 3a.

The z-component of the dipole moment increased steeply up to three-fold at higher electric field strengths reaching a plateau within the first 1 ns of the simulation and remaining constant throughout the rest of the simulation time. Due to the anisotropy of the CNF structure, its length axis is oriented in the electric field direction. A dipole moment of 214 Debye was noted for the optimized initial CNF structure. As shown in Figure 1a, the CNF model was initially offset by 10° from the y-axis.

Therefore, the characteristic of the y-component of the dipole moment was investigated as shown in Figure 3b. The results show that, as the CNF model rotated from the initial offset angle (10° from the y-axis) to align parallel to the electric field direction, a drop in the y-component of the dipole moment was observed. Figure 3b shows that the y-component of the dipole moment began to drop simultaneously as the onset of alignment, as shown in Figure 2a, reaches saturation when complete alignment is achieved and plateaus for the time when the alignment is maintained as shown in Figure 2b.

Additionally, a change in the y-component of the dipole moment is seen when the alignment is lost. This observation of the y-component of the dipole moment correlates with Table 1 and Figure 2. Except for 0.09 and 0.2 mV/nm, the x-component of the dipole moment showed no significant fluctuations for all the applied electric field strengths. This indicates no rotations and twisting of the CNF along its thickness direction for most of the simulation time.

However, towards the end of the simulation, a change in the x-component of the dipole was observed, similar to the mild twisting of the CNF reported in Table 1. The y- and x-components of the dipole moment for 0.09 mV/nm and 0.2 mV/nm did not follow the same trend as the other electric field strengths. Similarly, these two electric field strengths were characterized by rotation of the CNF and showed no alignment as shown in Table 1.

#### 3.1.3. Effect of Electric Field on the Cellulose Structure

While the results indicate induced CNF alignment at a low electric field, it is fundamental to understand the effect of the applied electric field on the cellulose structure. The CNF’s RMSD, the radius of gyration, end-to-end length, and hydrogen bond population were characterized. The analysis results indicate that all the electric field strengths, including the highest (0.2 mV/nm), have no discernible effect on the cellulose structure. RMSD quantitatively predicts the conformational stability of a simulation structure throughout the simulation. The deviation imposed by external stresses is calculated by comparing the simulated structure during simulation with the reference. RMSD is defined by Equation (1) [21].
(1)$$RMSD=\sqrt{\frac{1}{N}\overset{N}{\underset{i=1}{\sum }}{\left|r_{final}\left(i\right)-r_{initial}\left(i\right)\right|}^{2}}$$

Here, *r_final_*(*i*) and *r_initial_*(*i*) are the coordinates of an atom *i* in its final and the initial structure, respectively, and *N* is the number of atoms in the structure.

Appendix A shows the average value of the RMSD for each applied electric field strength over the entire simulation time. The analysis of the applied electric field on the molecular structure of the CNF model is shown in Appendix A. As can be seen, the RMSD value for all the simulations is less than 0.112 nm (Appendix A). All of the applied electric field strengths did not significantly disrupt the cellulose structure during the simulation. The RMSD values remained similar for most of the electric field strengths used. However, at higher electric fields, 0.09 and 0.2 mV/nm, characterized by rotations, a small increase in the RMSD value was observed.

The radius of gyration (*Rg*) quantifies the changes in shape and size of the simulation structure under external stresses (i.e., the electric field) during the MD simulations. *Rg* is the distribution of the atoms in space relative to their center of mass and can be defined using Equation (2) [21].
(2)$$Rg=\sqrt{\frac{1}{N}\overset{N}{\underset{i=1}{\sum }}{\left|r\left(i\right)-r_{center}\right|}^{2}}$$

Here, *r*(*i*) is the coordinates of an atom *i*, *r_center_* is the center of mass coordinates and *N* is the number of atoms in the structure.

Appendix A shows that all the electric field simulations produced compact structures similar to the zero-field simulation (*Rg*: 3.178 nm). Variations in the end-to-end distance of the cellulose structure were evaluated by determining the distance between the first atom and the last atom in the index group of the cellulose model [21]. Appendix A indicate that the electric field hardly affects the cellulose end-to-end distance (no elongation) in the z-direction as the electric field varies. Longer simulations were conducted to verify the phenomenon indicate similar *Rg* and end-to-end length observations as shown in Appendix A.

The hydrogen bond network in crystalline cellulose 1β plays a significant role in stabilizing the cellulose structure. A comprehensive analysis of the hydrogen bonds for all applied electric fields was conducted and compared to a zero-field simulation to better understand the effects of the electric field in the cellulose structure. The intrachain (O3H3…..O5 and O2H2…..O6’) and the interchain (O6H6…..O3 and O6H6…..O2) hydrogen bonds for all the applied electric fields were characterized. The overall intersheet hydrogen bond population was also calculated. In this study, the existence of hydrogen bonds is analyzed based on two geometric criteria: the donor-acceptor distance, *r* ≤ *r_HB_* = 0.35 nm and donor-acceptor angle, *α* ≤ *α_HB_* = 30°.

Figure 4 shows the evolution of the hydrogen bond population over the simulation time for various hydrogen bond types for all applied electric fields. As can be seen, the evolution of hydrogen bonds in all-electric fields follows a similar trend throughout the simulation time. The standard deviation of the averaged number of hydrogen bonds for all applied electric fields was found to be 1.757, 1.964, 3.749, and 1.731 for intrachain O3H3…..O5, intrachain O2H2…..O6’, interchain O6H6…..O3, and interchain O6H6…..O2, respectively, and they were all dominantly present everywhere in the cellulose structure.

The cellulose hydrogen-bonding networks within the cellulose structure were not significantly affected by the applied low electric fields. A closer look at the hydrogen bonds evolution trends reveals that within the first nanosecond into the simulation, the number of intrachain O3H3…..O5 (Figure 4a), intrachain O2H2…..O6’ (Figure 4b), interchain O6H6…..O3 (Figure 4c), and interchain O6H6…..O2 (Figure 4d) hydrogen bonds falls and stabilizes to mean values ~101, 226, 60, and 159, respectively, throughout the remaining simulation time. This trend was observed in all simulations.

The repeated simulation for an extended time yields similar results for the intrachain, interchain, and overall hydrogen bonds as shown in Appendix A. The fall is attributed to the relaxation of the cellulose structure at the onset of cellulose [19,34,35]. Due to the strong hydrogen bonding networks in cellulose and the heavy hydroxyl groups within the cellulose structure, the demolished hydrogen bonds can reconstruct new hydrogen bonds with the nearest donor or acceptor atom.

As a result, the evolution of intersheet hydrogen bonds with the O6H6…..O2 donor-acceptor atoms was investigated. The results revealed a gradual increase of hydrogen bonds within the first nanosecond from 154 to an average of 295 for all applied electric fields as shown in Appendix A. This could indicate the formation of new intersheet hydrogen bonds due to the disruption of the intrachain and interchain hydrogen bonds.

Analyzing distance distributions of these hydrogen bonds (Figure 5) shows that the intrachain O3H3…..O5 hydrogen bond is the strongest with a donor-acceptor distance of 0.2725 nm with an occupancy ~36% for no field and all-electric fields. The interchain O6H6…..O3 hydrogen bond has a donor-acceptor distance of 0.2775 nm, and both intrachain O3H3…..O5 and interchain O6H6….O3 hydrogen bonds show a narrow distribution distance indicating that their occurrence is within the peak values. Both intrachain O2H2…..O6’ and interchain O6H6…..O2 hydrogen bonds show a donor-acceptor distance of 0.2875 nm.

Based on the criteria used in the analysis, the hydrogen bond distance distributions for intersheet hydrogen bonds (Appendix A) indicate that they are the weakest. Furthermore, the hydrogen bond donor-acceptor distance of the intersheet bonds increases with an increase in the electric field from 0.2925 to 0.3375 nm for no field and 0.2 mV/nm, respectively, indicating much weaker intersheet hydrogen bonds at high electric field strength. The donor-acceptor distances and occupancy for all the applied electric fields are summarized in Table 2. The hydrogen bond lengths remained constant for all applied electric fields. A similar observation is seen at a similar electric field (0.06 mV/nm) as shown in Appendix A. This conclusively implies that CNFs can be aligned in low electric fields with negligible cellulose structure effects.

### 3.2. CASE 2: Electric Field along the X-Axis

#### 3.2.1. Alignment Effect of Electric Field on Cellulose

This study focused on the effect on CNF if an electric field is applied in the x-direction, cutting across the thickness direction of the cellulose structure as shown in Figure 6. The CNF model was titled 45° from the original Cartesian x-axis, and the external electric field was applied as described in Figure 1b. Table 3 represents snapshots of the CNF alignment under the applied electric field direction along the x-axis.

When an electric field is applied, the CNF chains are supposed to be aligned parallel to the electric field. However, due to relatively solid cellulose chain networks, cellulose chains cannot be aligned along the film thickness direction. Yet, OH groups are still sensitive to the electric field, which results in the rotation of the cellulose chains in the out-of-plane direction such that more OH groups appear on the outside of the structure [18]. The results of this setup indicate a two-part orientation phenomenon.

First, a gradual clockwise rotation of the CNF from its initial orientation (45°) to align with the Cartesian x-axis due to the rotation of CNF chains in the out-of-plane direction was observed. Second, once the CNF is oriented in the Cartesian x-axis, expectedly, a slow rotation of the CNF follows to align its principal axis (z-axis) parallel to the direction of the applied electric field (x-axis). The primary goal of this simulation (CASE 2) is to observe the alignment of CNF, as has been reported in the literature, when an electric field is applied across the CNF’s thickness direction. Therefore, in this case, the scope of analysis was limited to the first part of the observed orientation as shown in Figure 7.

The angle of orientation (tilt) of the CNF cross-section about the original Cartesian x-axis was used to estimate the alignment induced by the applied electric field in this case. The CNF model’s front and middle cross-sections were used in the calculation to ascertain behavioral correlation throughout the model’s length. The front cross-section and the middle cross-section of the CNF model at each nanosecond were analyzed to deduce the alignment by measuring the degree to which the cellulose x-axis orients about the x-axis as shown in Figure 7.

The cellulose alignment and field strength dependence are readily seen. Upon applying the electric field for 1 mV/nm, cellulose’s x-axis remained stable at the initial orientation (45°) for approximately 2 ns before gradually rotating clockwise to align with the Cartesian x-axis. However, this electric field strength was insufficient to align the cellulose’s x-axis completely and could only achieve a degree of orientation of approximately 13° about the Cartesian x-axis at 6 ns as shown in Figure 7.

This was followed by a slow rotation of the CNF model to parallel its principal axis to the applied electric field. A two-fold increase of the applied electric field strength to 2 mV/nm, as shown in Figure 7, resulted in the complete orientation of cellulose’s x-axis about the Cartesian x-axis at 15 ns of simulation before rotating to align parallel to the applied electric field began.

This observation coincides with the experimental result [18]. Increasing the electric field strength further to 4.5 mV/nm showed a similar trend; however, the complete orientation of cellulose’s x-axis about the Cartesian x-axis was achieved at 4 ns earlier than 2 mV/nm. Furthermore, the degree of orientation of cellulose’s x-axis to the Cartesian x-axis was more gradual for 2 mV/nm than observed at 4.5 mV/nm. Both the 2 and 4.5 mV/nm electric fields stably maintained the initial degree of orientation (45°) for approximately 1 ns, followed by slight instability before the onset of the orientation.

This could be due to the corresponding resistance of the CNF model to the high external force imposed by the applied electric field. Similarly to 1 mV/nm, after the CNF model rotated out-of-plane to align its x-axis to the Cartesian x-axis, a slow rotation to align the CNF model’s principal axis parallel to the applied electric field (x-axis) followed.

#### 3.2.2. Dipole Moment

As shown in Figure 7 and Table 3, gradual alignment of the CNF (initially offset at 45°) is observed to reorient it back to the Cartesian x-axis. Figure 8a shows the x-component of the total dipole moment. It is worth noting that the dipole moment profile has similar characteristics to the angle of orientation shown in Figure 7. The CNF initially resisted the applied electric field and remained stable for 2 ns before it began to rotate for alignment. During this time, the x-component of the dipole moment increased steeply while the CNF maintained the original orientation angle (45°).

At 2 and 4.5 mV/nm, a characteristic jump in the orientation angle is observed before orientation begins. The dipole moment magnitude (x-component) increases as the electric field intensity increases. As shown in Figure 7, the CNF cross-section gradually rotates clockwise towards the Cartesian x-axis. During the gradual alignment phase, a gradual drop in the magnitude of the dipole is observed with saturation after 6, 16.26, and 12.29 ns for the 1, 2, and 4.5 mV/nm cases, respectively.

These results of the x-component of the dipole moment (Figure 8a) correspond to the angle of orientation saturations and snapshots shown in Figure 7 and Table 3. This drop can be ascribed to the dipole’s translation in the electric field’s direction as the CNF rotates clockwise. The cellulose chains in the CNF model are aligned along the z-direction (chain length). The obvious expectation is that an object will rotate to align along the direction of the applied electric field. However, due to the cellulose chains’ strong interactions, they cannot be aligned in the thickness direction (x-direction as in this case).

Interestingly, when exposed to an electric field along the thickness direction, the OH groups of cellulose tend to align in the out-plane direction [18]. As previously stated, a two-part orientation phenomenon was observed in which the CNF first rotated clockwise from the initial angle of orientation (45°) to align with the Cartesian x-axis, followed by the expected behavior of aligning its principal axis parallel to the applied EF. The tendency of cellulose’s OH groups to align in the out-of-plane direction was observed in the first orientation phenomenon, with evident coherence in the angle of orientation (Figure 7) and the x-component of dipole moment (Figure 8a).

It is reasonable to say that this phenomenon can be useful in tuning the surface properties of cellulose, such as the water contact angle, using an electric field. After the CNF was oriented in the Cartesian x-axis, the second part of orientation began where the CNF rotated to align its principal axis parallel to the applied EF. This second orientation phenomenon is evident in Figure 8a, where the x-component of the dipole moment changes course as the CNF rotates in the direction of the applied electric field. The z and y components of the dipole moment were also investigated as shown in Figure 8b,c, respectively.

The CNF’s chain length was oriented on the z-axis, while the electric field was applied across the x-axis. During the gradual rotation of the CNF from the initial angle of orientation (45°) to the Cartesian x-axis, the z-component of the dipole moment (Figure 8b) shows no fluctuations for the applied electric field strengths indicating no rotation or twisting in the CNF’s chain length.

The y-component of the dipole moment (Figure 8b) exhibits negative (directional) values and reaches saturation as the CNF aligns with the Cartesian x-axis, coherent with Figure 7 and Figure 8a. The z-component of the dipole moment (Figure 8b) shows significant changes due to the rotation of the CNF’s length axis during the second part of the rotation to align the CNF’s chain length (z-axis) parallel to the direction of the electric field (x-direction), whereas the y-component of the dipole moment (Figure 8c) shows no fluctuations.

#### 3.2.3. Stability of Cellulose Structure

As in the previous case, the stability of the cellulose structure in the high electric fields applied for this case was investigated by evaluating the RMSD, the radius of gyration, the end-to-end length, and the overall evolution of hydrogen bonds as summarized in Appendix A. The simulation results indicate that the electric field induced no structural changes in the CNF model. Appendix A shows the RMSD evolution of the cellulose structure under the influence of all the applied electric fields (x-axis).

The RMSD value remained stable but increased to 0.1124 from 0.1039 nm (no field) when an external electric field of 1 mV/nm was applied across the x-axis. The RMSD value increased to 0.1280 nm with the increasing electric field at 4.5 mV/nm. The deviation in the RMSD value remained small, below 0.13 nm, which indicates that the high electric fields used in this case are not high enough to change the cellulose structure. This could be attributed to the rotation of the cellulose structure as reported in Figure 7.

Applying high electric fields across the x-axis showed that the cellulose cross-section hydrogen bonds align along the applied electric field. The radius of gyration did not show any deviation and remained at 3.178 nm for all the applied field strengths (Appendix A). Evaluation of the end-to-end distance showed no deviation from the cellulose chain length in the absence of an electric field (Appendix A). The hydrogen bond evolution when cellulose was subjected to high electric fields was further analyzed.

In Appendix A, the average number of the overall hydrogen bonds indicates a small standard deviation (3.851) of hydrogen bonds for all the applied electric fields. Investigating the donor-acceptor distance distribution showed similar values of 0.288 nm and the same occupancy at all-electric fields as shown in Appendix A.

## 4. Conclusions

The alignment of the crystalline cellulose nanofiber model under external electric fields was investigated in this study as well as the subsequent effects on structural stability. The CNF model was crystalline cellulose 1β consisting of 18 chains in an aqueous environment and room temperature, and GROMACS software was used for the MD simulation. The induced alignment of CNF along the direction of the applied electric fields was investigated, and the changes in the physical, structural, and dynamic properties of the crystalline CNF model were inspected under the applied electric fields. The electric field was applied along the CNF length and the transverse directions. The RMSD, radius of gyration, end-to-end length, and overall hydrogen bonds were investigated to examine the structural denaturing due to the electric field.

Under low electric fields, cellulose chains can reorient and align in the direction of the applied electric field with no effect on the cellulose structure. The low electric fields can maintain the achieved alignment. The present model and study revealed that cellulose alignment could be achieved by applying electric fields without denaturing its inherent structural properties. The insights gained through this study are useful in exploring the design of electric-field-induced processing techniques for cellulose filaments, thin films, and electro-active cellulose composites.

## Figures and Tables

**Figure 1 polymers-14-01925-f001:**
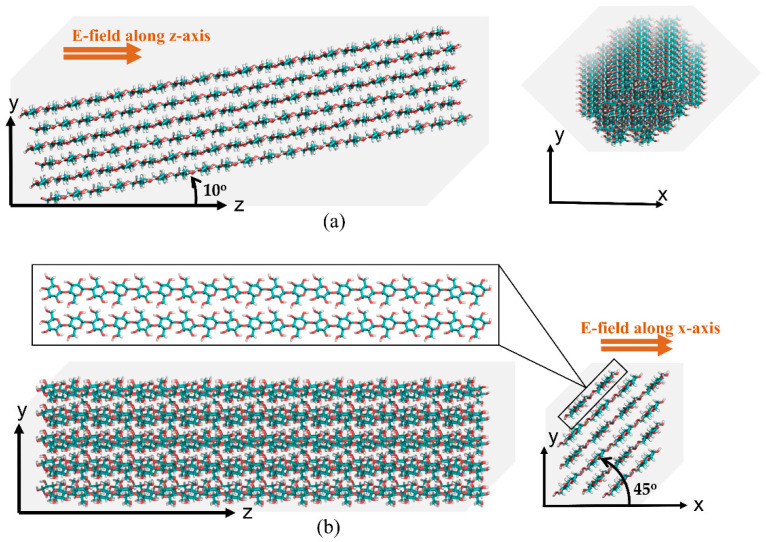
Electric field cases on the CNF model: (**a**) along the length direction (CASE 1) and (**b**) along the transverse direction (CASE 2).

**Figure 2 polymers-14-01925-f002:**
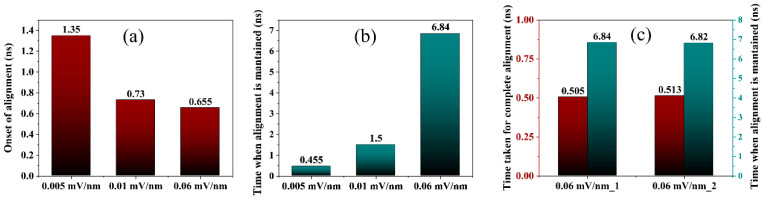
(**a**) Onset of cellulose alignment in the electric field, (**b**) the time when alignment in the electric field is maintained, and (**c**) the time taken to complete and maintain alignment at 0.06 mV/nm.

**Figure 3 polymers-14-01925-f003:**
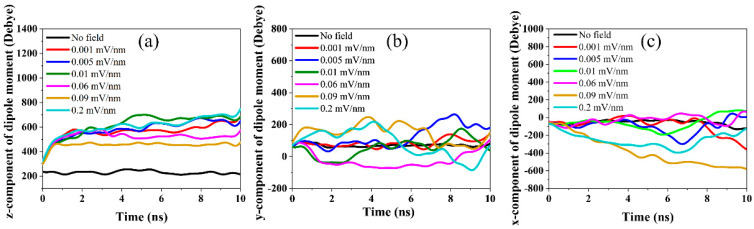
The (**a**) z-component, (**b**) y-component, and (**c**) x-component of the dipole moment of CNF exposed in the different electric field strengths along the z-direction.

**Figure 4 polymers-14-01925-f004:**
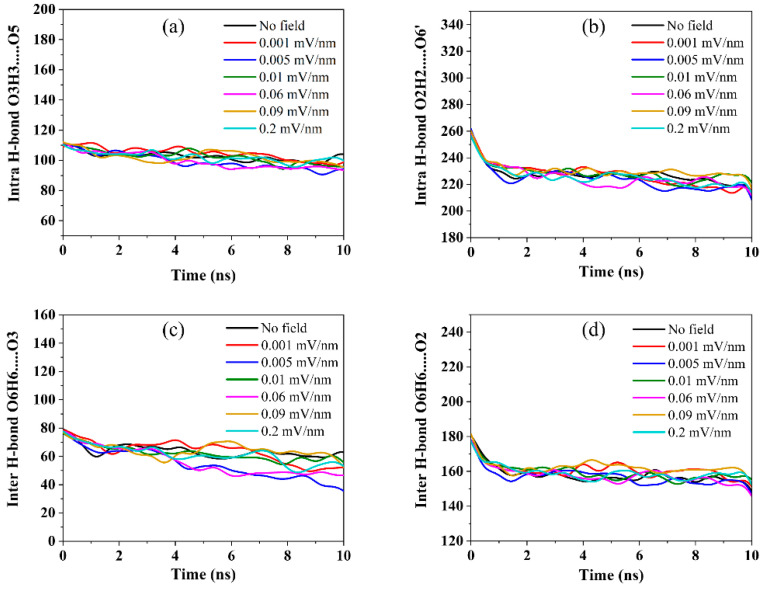
Hydrogen bond population (electric field along the z-axis) (**a**) intrachain O3H3…..O5, (**b**) intrachain O2H2…..O6’, (**c**) interchain O6H6…..O3 and (**d**) interchain O6H6….O2.

**Figure 5 polymers-14-01925-f005:**
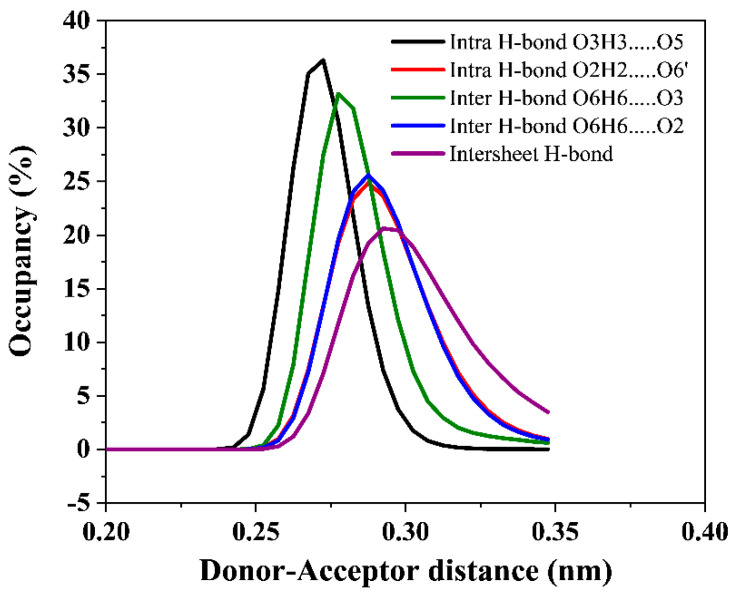
The donor-acceptor distance and occupancy of hydrogen bonds with an electric field along the z-axis.

**Figure 6 polymers-14-01925-f006:**
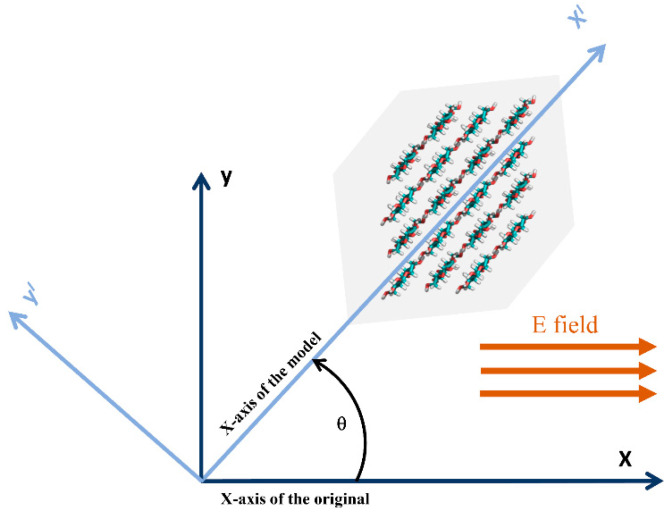
Setup of high electric field strength applied along the x-axis: the angle *θ* shows the tilt reference used for the analysis.

**Figure 7 polymers-14-01925-f007:**
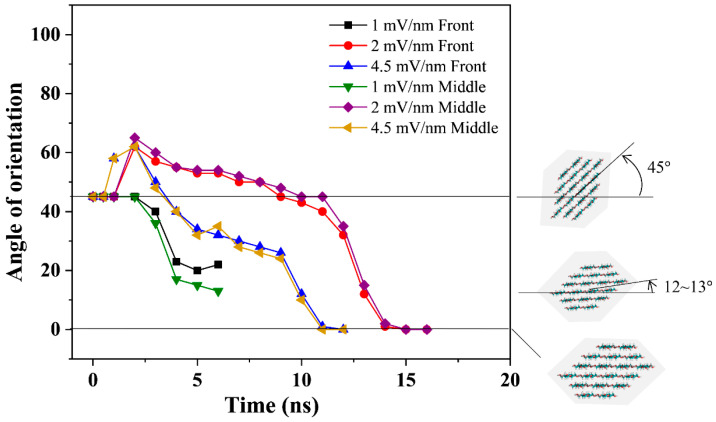
The orientation angles of the front and middle cross-sections of the CNF model to the x-axis throughout the simulation time for all the applied electric field strengths.

**Figure 8 polymers-14-01925-f008:**
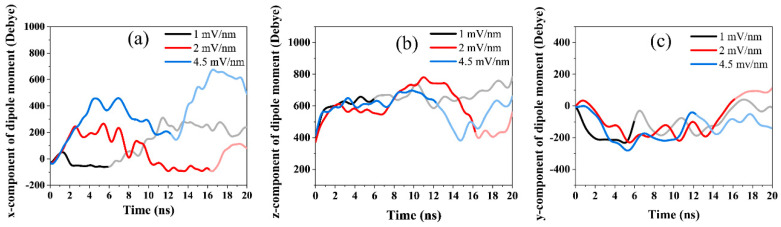
The x-component (**a**), z-component (**b**), and y-component (**c**) of the dipole moment of CNF exposed to various electric field strengths along the x-direction. (The first part of the orientation to align the CNF from the initial angle to the Cartesian x-axis is denoted by bright black, red, and blue colors. The faded black, red, and blue denote the second part of the CNF’s orientation to align its chain length parallel to the EF’s direction).

**Table 1 polymers-14-01925-t001:** Snapshots of CNF alignment along with the applied external electric fields (z-axis).

Electric Field (mV/nm)	0	1 ns	2 ns	3 ns	4 ns	5 ns	6 ns	7 ns	8 ns	9 ns	10 ns
**0.001**	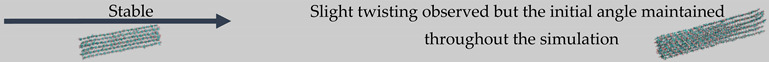
**0.005**	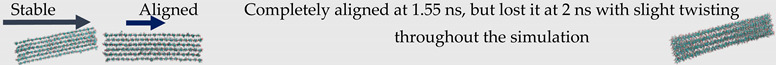
**0.01**	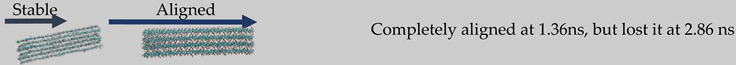
**0.06**	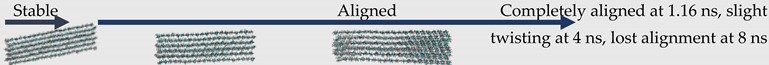
**0.09**	Characterized by rotation
**0.2**	Characterized by rotation

**Table 2 polymers-14-01925-t002:** The hydrogen bond formation distance between the donor and acceptor atoms of cellulose and their occupancy under an external electric field (electric field along the z-axis).

	Intra H-Bond O3H3…..O5		Intra H-Bond O2H2…..O6’		Inter H-Bond O6H6…..O3		Inter H-Bond O6H6…..O2		Intersheet H-Bond	
Electric Field (mV/nm)	Bond length (nm)	Occupancy (%)	Bond length (nm)	Occupancy (%)	Bond length (nm)	Occupancy (%)	Bond length (nm)	Occupancy (%)	Bond length (nm)	Occupancy (%)
No field	0.2725	36.32	0.2875	24.86	0.2775	33.18	0.2875	25.61	0.2925	20.62
0.001	0.2725	36.22	0.2875	24.88	0.2775	32.47	0.2875	25.34	0.2925	20.13
0.005	0.2725	36.09	0.2875	24.61	0.2775	32.82	0.2875	25.18	0.2925	19.65
0.01	0.2725	36.43	0.2875	24.79	0.2775	32.81	0.2875	25.37	0.2925	20.33
0.06	0.2725	36.17	0.2875	24.79	0.2775	33.06	0.2875	25.39	0.2975	19.94
0.09	0.2725	36.38	0.2875	24.93	0.2775	33.64	0.2875	25.76	0.2975	20.55
0.2	0.2725	36.16	0.2875	24.66	0.2775	32.37	0.2875	25.26	0.3375	29.59

**Table 3 polymers-14-01925-t003:** Snapshots of CNF alignment and the applied external electric fields in the x-axis.

0 ns	2 ns	4 ns	6 ns	8 ns	10 ns	12 ns	14 ns	16 ns	18 ns	20 ns
**1 mV/nm**
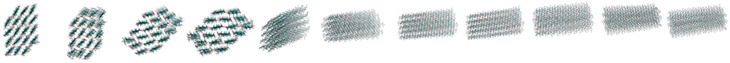
**2 mV/nm**
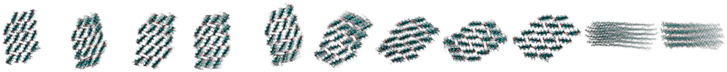
**4.5 mV/nm**
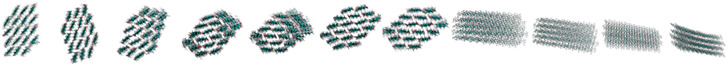

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
