# Peer review of "Molecular Dynamics Study of Cellulose Nanofiber Alignment under an Electric Field"

_polymers, 2022, doi:10.3390/polym14091925_

Round 1

Reviewer 1 Report

The authors performed detailed atomistic simulation study on the cellulose nano fiber (CNF) under external electric field with different crystal orientations and different electric field strengths.  By analyzing the dynamics of the CNF under the electric field combined with structural analysis, including the alignment time, dipole moment,  RMSD, radius of gyration and the end-to-end distance analysis, they found that under low strength electric field, the structure of the CNF is basically stable and the alignment of the crystal could be controlled by the electric field. They also found the source of the alignment could come from the -OH groups on the cellulose chain which drive the whole crystal move/rotation under the external electric field. The study was carried out in very detailed level and the finding of the CNF crystal alignment to the electric field was found match the experimental observations. The paper was written well, and the logic of the whole research is clear and well presented.  I would like to recommend this paper to be published in Polymers after the authors fixed the following issues:

  1. Please add the detailed information about the interaction parameters of the particles used in the system so others could reproduce your results. The authors could list the interaction parameters/particle types in supporting information documents to avoid additional modification to the main text.
  2. Please add system overview table to include all system components details (such as the number of the TIP3P water, simulation box type/dimension etc).
  3. The snapshots shown in fig 1 looks like the boundary of the simulation box are close to the edge of the CNF. Is it only an cropped snapshot or the whole simulation box looks like that? I would like to see at least one true snapshot of the system that shows the crystal and water together.
  4. In Table 1, the authors observed a rotation movement when the CNF is under stronger electric field. Here, I am a little bit confused about how this rotation was generated. Could the author provide additional discussion about the reason why the CNF rotates? Are these rotations just temporary oscillation and eventually stopped or it will rotate forever? It is weird to see a motor like motion could happen to a CNF under constant electric field. Is it an artifact?
  5. The authors should include all equations such as calculation method of the whole CNF dipole moment, Rg calculation equation, end-to-end distance calculation equation into the main text.
  6. The authors argue that the OH group could affect the rotation of the whole CNF but didn’t go very deep. The authors should add one plot about how the orientation of the OH group (O-H bond vector) respond to the dynamics of the whole CNF system as a proof to show if the OH group play a very important role in CNF rotation movements.

Reviewer 2 Report

The authors present a computational model of the alignment of cellulose nanofibers in electric fields.  The paper is well written and presents a clearly designed and carried out computational model and reveals several interesting findings including subtle and somewhat non-intuitive differences in alignment and molecular behaviors in fields of different strengths.  Furthermore, the authors were after to verify previous experimental results regarding nanocellulose alignment, which strengthened this paper.  Overall, the paper represents a excellent contribution to the field and provides a solid platform to extend into other models.  In the larger pictures such modeling will enable more careful experimentation approach to manipulating and controlling nanoscale cellulose organization.

Author Response

Thank you for the comments.